



# Enhanced Mediterranean water cycle explains increased humidity during MIS 3 in North Africa

Mike Rogerson[1]

Yuri Dublyansky[2]

Dirk L. Hoffmann[3]

Marc Luetscher[2,4]

Christoph Spötl[2]

Paul Töchterle[2]

School of Environmental Sciences, University of Hull, Cottingham Road, Hull, HU6 7RX, UK.
Institute of Geology, University of Innsbruck, Innrain 52, 6020 Innsbruck, Austria.
Department of Human Evolution, Max Planck Institute for Evolutionary Anthropology, Deutscher

Platz 6, 04103, Leipzig, Germany

Swiss Institute for Speleology and Karst Studies (ISSKA), Serre 68, CH-2300 La Chaux-de-Fonds





## Abstract

*We report a new fluid inclusion dataset from Northeast Libyan speleothem SC-06-01, which is the*

*largest speleothem fluid inclusion dataset for North Africa to date. The stalagmite was sampled in*

*Susah cave, a low altitude coastal site, in Cyrenaica, on the northern slope of the Jebel Al-Akhdar.*

*Speleothem fluid inclusions from latest Marine Isotope Stage (MIS) 4 and throughout MIS 3 (~67 to*

*~30 ka BP) confirm the hypothesis that past humid periods in this region reflect westerly rainfall*

*advected through the Atlantic storm track. However, most of this moisture was sourced from the*

*Western Mediterranean, with little direct admixture of water evaporated from the Atlantic. Moreover,*

*we identify a second moisture source likely associated with enhanced convective rainfall within the*

*Eastern Mediterranean. The relative importance of the western and eastern moisture sources seems to*

*differ between the humid phases recorded in SC-06-01. During humid phases forced by precession,*

*fluid inclusions record compositions consistent with both sources, but the 52.5 – 50.5 ka interval*

*forced by obliquity reveals only a western source. This is a key result, showing that although the*

*amount of atmospheric moisture advections changes, the structure of the atmospheric circulation over*

*the Mediterranean does not fundamentally change during orbital cycles. Consequently, an arid belt*

*must have been retained between the Intertropical Convergence Zone and the mid-latitude winter*

*storm corridor during MIS 3 pluvials.*

## Introduction

Atmospheric latent heat is a major component of global and regional climate energy budgets and

changes in its amount and distribution are key aspects of the climate system (Pascale et al., 2011).

Equally, in mid- and low-latitude regions, changes in the water cycle have more impact on landscapes

and ecosystems than changes in sensible heat (Black et al., 2010). Rainfall in semi-arid regions is thus

one of the key climate parameters that understanding future impact on human societies depends upon

(IPCC, 2014), making constraining of mid-latitude hydrology a globally significant research priority.

These regions, however, have a particularly sparse record of palaeoclimate due to typically poor

preservation of surface sedimentary archives (Swezey, 2001). North Africa is a region that fully



exhibits these limitations, and large areas present either no pre-Holocene record or else they present
highly discontinuous deposits indicating major reorganisation of the hydroclimate, which are
challenging to date (Armitage et al., 2007). North Africa also fully exhibits the progress
palaeoclimatologists have made in understanding continental hydrological change from its impact on
the marine system; our understanding of past North African hydroclimate is disproportionately drawn
from records from the Mediterranean Sea (Rohling et al., 2015) and the eastern Central Atlantic
(Goldsmith et al., 2017;deMenocal et al., 2000.;Adkins et al., 2006).

**Past changes in North African hydroclimate**
Marine-based evidence offers a coherent model in which changes in the spatial distribution of
insolation alter atmospheric circulation on orbital timescales ($10^4$ to $10^5$ years) and force major
reorganisations of rainfall in semi-arid regions such as the Sahel and southern Saharan regions
(Rohling et al., 2015;Goldsmith et al., 2017). This result is at least partially confirmed in climate
modelling experiments (Bosmans et al., 2015;Tuenter et al., 2003) and provides a conceptual
framework in which fragmentary evidence of hydrological change on the adjacent continent can be
understood (Rowan et al., 2000). There is 1) strong geochemical evidence that runoff from the
African margin initiated the well-known "sapropel" thermohaline crises of the eastern Mediterranean
(Osborne et al., 2010;Osborne et al., 2008) and, 2) convincing evidence that the southern margin of
the Mediterranean was more variable than the northern in terms of the relative magnitude of
precipitation changes and the distribution of flora, fauna and hominid populations (Drake et al., 2011).
However, we emphasise the fact that this understanding is largely drawn from evidence from outside
continental North Africa, and that this limits our knowledge about the nature and impact of
hydrological changes in this region.
There is strong evidence for a more humid climate throughout the Sahara and Sahel regions during the
early Holocene (Gasse and Campo, 1994;Gasse, 2002;Fontes and Gasse, 1991;Prentice and Jolly,
2000;Jolly et al., 1998;Collins et al., 2017), and in older interglacial periods (Drake et al.,



2008;Armitage et al., 2007;Vaks et al., 2013). This evidence has been interpreted to indicate that
humid conditions extended from the modern Sahel (~15ºN) to the Mediterranean coast (30-35ºN).
However, this only partially agrees with model results, which do indicate orbitally forced migration of
the monsoon belt but not across such a large spatial scale as suggested by the empirical data. Model
experiments indicate that monsoonal rainfall occurring within the Intertropical Convergence Zone
(ITCZ) likely extended no further north than ~23ºN (Harrison et al., 2015). This well-recognised lack
of agreement between rainfall fields in model experiments for the past and reconstructed
hydrographies from the distribution of lakes and vegetation (via pollen) (Peyron et al., 2006) remains
an major research problem. While some models also suggest that during times of high Northern
Hemisphere insolation, enhanced westerlies advected Atlantic moisture into the basin (Brayshaw et
al., 2009;Tuenter et al., 2003;Bosmans et al., 2015), high-resolution regional modelling indicates that
this primarily affected the northern Mediterranean margin only (Brayshaw et al., 2009). This result is
consistent with evidence of enhanced runoff at these times from the southern margin of Europe
(Toucanne et al., 2015). On the African coast east of Algeria, the southern limit of enhanced
precipitation arising from increased westerly activity within model experiments essentially lies at the
coastline (~32ºN), and does not appear to drive terrestrial hydrological changes. Overall, there is
therefore a striking mismatch between the apparent humidity of Africa between 23 and 32ºN in the
empirical record (a zonally oriented belt ~1000 km in width) and the climate models. This region
encompasses southern Tunisia, in which multiple lines of evidence for distinct and widespread
periods of increased humidity provide a highly secure basis for enhanced rainfall during Northern
Hemisphere insolation maxima (Ballais, 1991;PETIT-MAIRE et al., 1991), the Fezzan basin, in
which compelling evidence for multiple lake highstands exists (Drake et al., 2011) and western Egypt,
where large tufa deposits attest to higher past groundwater tables (Smith et al., 2004).
It is unlikely that significant further progress will be made in understanding the palaeoclimate of
North Africa without new empirical evidence of regional hydrological changes from which
atmospheric dynamics can be delineated.



**The central North African speleothem record**
Speleothem palaeoclimatology has high potential for North Africa, but is only recently becoming
established through key records developed for Morocco (Wassenburg et al., 2013;Ait Brahim et al.,
2017;Wassenburg et al., 2016). Until recently, the only speleothem record published from central
North Africa was a single continuous record from 20 to 6 ka BP from northern Tunisia (Grotte de la
Mine). This record shows a large deglacial transition in both $\delta^{13}C$ and $\delta^{18}O$ (Genty et al., 2006), with
oxygen isotopes indicating a 2-step change from a relatively isotopically heavy (-5‰) LGM (20-16 ka
BP), through an intermediate (-6 to -7‰) deglacial period (16-11.5 ka BP) to a relatively isotopically
light early Holocene. The $\delta^{13}C$ record indicates cool periods exhibiting higher carbon isotope values,
more clearly delineating the Bølling-Allerød / Younger Dryas oscillation than $\delta^{18}O$. This is assumed
to reflect higher soil respiration during warm periods (Genty et al., 2006). A major change in the
carbon isotopic composition occurred across the transition from the relatively arid glacial to the more
humid Early Holocene, and indicates a significant reorganisation of the regional hydroclimate.
However, it is difficult to interpret these data in isolation. A recently reported speleothem record (SC-
06-01) indicates that conditions in northern Libya during Marine Isotope Stage 3 (MIS 3) were more
humid than today, and shows isotopic evidence of a teleconnection between temperature in Greenland
and rainfall at the southern Mediterranean margin (Hoffmann et al., 2016). The oxygen isotope record
indicates that the water dripping into the cave during MIS 3 was isotopically too heavy for the
moisture to be sourced from within the monsoon system (Hoffmann et al., 2016). However, beyond
ruling out a southern source $\delta^{18}O_{cc}$ values alone are not sufficient to determine the origin of
atmospheric vapour. Three distinct humid phases within MIS3 are reported from this speleothem: 65-
61 ka, 52.5-50.5 ka and 37.5-33 ka. Phases I and III occur during times of low precession, when
summer insolation on the northern hemisphere is relatively increased. Phase II represents the first
evidence for high obliquity being able to cause a pluvial period in the north African subtropics in the
same manner as precession (Hoffmann et al., 2016). In SC06-01, all three growth phases are fractured
into multiple short periods of growth, and show a marked temporal coherence with Greenland



Dansgaard-Oeschger interstadials (Hoffmann et al., 2016). Here, we report fluid inclusion data from
this speleothem and discuss how this helps resolve some of the issues discussed above.
Speleothem fluid inclusions are small volumes of water that were enclosed between or within calcite
crystals as they grew, ranging in size from less than 1 μm to hundreds of μm (Schwarcz et al., 1976).
This water represents quantities of ancient drip-water that can be interrogated directly to ascertain the
isotopic properties of the oxygen ($\delta^{18}O_{fi}$) and hydrogen ($\delta^2H_{fi}$) it comprises. This powerful approach
circumvents some of the uncertainty inherent in the interpretation of the stable isotopic values
preserved in the calcite comprising the speleothem itself ($\delta^{18}O_{cc}$, $\delta^{13}C_{cc}$). Fluid inclusion isotopes have
been used to demonstrate changes in air temperatures (Wainer et al., 2011;Meckler et al.,
2015;Arienzo et al., 2015) and in the origin of the moisture from which precipitation was sourced
(McGarry et al., 2004;Van Breukelen et al., 2008). Fluid inclusions from speleothems in Oman have
also been used to identify monsoon-sourced precipitation during interglacial phases (Fleitmann et al.,
2003), providing a rationale for similar investigation of fluid inclusion isotope behaviour in North
Africa.
Material and Methods
SC-06-01 is a 93-cm long stalagmite from Susah Cave (Fig. 1, 32º53.419' N, 21º52.485' E), which
lies on a steep slope ~200 m above sea level in the Al Akhdar massif in Cyrenaica, Libya (Fig. 1). The
region is semi-arid today, with mean annual temperature ~20ºC and receiving less than 200 mm
precipitation per year, mostly in the winter (October to April). The Al Akhdar massif has thin soil
cover and a Mediterranean "maquis" vegetation. Susah Cave is hydrologically inactive today, and all
formations are covered with dust. The chronology of the speleothem and the general features of its
growth and $\delta^{18}O_{cc}$ record are published elsewhere (Hoffmann et al., 2016), and this study focuses on
fluid inclusion isotopes, their impact on the interpretation of $\delta^{18}O_{cc}$ and to a lesser extent on $\delta^{13}C_{cc}$ and
Sr isotopes.





Calcite isotopes were measured using a ThermoFisher Delta$^{plus}$XL isotope ratio mass spectrometer
(IRMS) equipped with a Gasbench II interface at the University of Innsbruck, according to standard
methods (Spötl, 2011). Fluid inclusions were examined in doubly-polished thick section (100 μm)
slides, using a Nikon Eclipse E400 POL microscope. The isotope composition of fluid inclusion water
was measured at the University of Innsbruck using a Delta V Advantage IRMS coupled to a Thermal
Combustion/Elemental Analyser and a ConFlow II interface (Thermo Fisher) using the line, crusher
and cryo-focussing cell described in Dublyansky and Spötl (2009). Samples were cut with a diamond
band saw along visible petrographic boundaries in the speleothem, and therefore represent specific
growth increments. Samples were analysed at least in duplicate, with the standard sampling protocol
used on the Innsbruck instrument (Dublyansky and Spötl, 2009). To exclude the possibility of post-
depositional diagenetic alteration, petrographic thin sections were investigated using transmitted-light
microscopy. Results are detailed in Supplemental Information 1.
Optical emission spectroscopy (OES) was used to measure a variety of elemental concentrations,
including Sr, along the main growth axis of SC-06-01. The low spatial resolution of trace elemental
analyses (every 10 mm) does not allow to investigate time series of elemental variation but was useful
to assess Sr contents of the samples for Sr isotope measurements by thermal ionisation mass
spectrometry (TIMS). The samples for TIMS analyses were drilled using a hand held micro drill with
a tungsten carbide drill bit. Sample sizes range between 2 and 4 mg, thus we achieved a minimum Sr
load of 100 ng on the Re filaments for TIMS. Chemical sample preparation and subsequent TIMS
measurement were done following standard protocols (Charlier et al., 2006). No spike was added to
the samples prior to chemical purification. The Sr isotope measurements were done on a Triton TIMS
housed at the Bristol Isotope Group laboratory, University of Bristol.



## Results

### Fluid inclusions

Petrographic analysis of the thick sections indicates that the distribution of fluid inclusions is highly

variable, with macroscopically opaque "milky" calcite typical of rapidly growing intervals containing

sometimes very abundant inclusions and the discoloured, translucent calcite of the slowly growing

intervals being almost inclusion-free (Fig. 2). In most samples, two distinct populations of inclusions

were identified with numerous small intra-crystalline inclusions and larger, but less frequent, inter-

crystalline inclusions. Consequently, the volume of water analysed per sample was very variable (Fig.

3). Indeed, a significant proportion of individual fluid inclusion measurements had analyte volumes

too small (<0.1 μL) to have confidence in the isotope results. A small number of analyses failed due

to excessive water saturating the detector, and these have not been included in the datasets presented

here. The major impact of the highly variable availability of inclusions in the speleothem is a

significant bias in the analyses towards the most rapidly growing, and therefore probably humid, time

periods.

In most samples, achieving within-error replication ($\delta^2H$ ±1.5‰, $\delta^{18}O$: ±0.5‰) of both $\delta^{18}O_{fi}$ and

$\delta^2H_{fi}$ was difficult. This must reflect more than one population of inclusions with different properties

being present within at least some samples, and each replicate analysis represents some proportion of

mixing between these populations. This suggests significant short-term variability in the composition

of the water stored in the presumably rather small soil/epikarst zone overlying the cave.

Consequently, any given time interval risks being under-sampled with regard to variability at that

time. Although there is some visual correspondence between the $\delta^{18}O_{fi}$, $\delta^2H_{fi}$ and $\delta^{18}O_{cc}$ data series

(Fig. 4), the usefulness of interpretation that can be drawn from the episodic SC-01-06 fluid inclusion

dataset when arranged as a time series is limited. We therefore largely limit our discussion to the

properties of the population of waters as a full dataset.

Figure 5 shows the SC-06-01 fluid inclusion dataset alongside Global Natural Isotopes in

Precipitation (GNIP) datasets from Tunis World Meteorological Office (WMO station 6071500), Sfax



(6075000) and Bet Dagan (4017900) (locations in Fig. 1) and other published precipitation datasets.
The Tunisian datasets fit within a trend typical of the Global Meteoric Water Line (GMWL) ($\delta^2$H =
$8\delta^{18}$O + 10). However, all this data lies along a single moisture evolution trend, and the Tunis and
Sfax populations overlap. The data from Bet Dagan exhibits a trend which is extremely close to being
parallel to the global trend dominating in Tunisia, but translated by +10 ‰ in $\delta^2$H, reflecting greater
deuterium excess. This is typical of the Mediterranean Meteoric Water Line (MMWL) (Ayalon et al.,
1998;Gat et al., 2003), and reflects internal recycling of water with consequent deuterium enrichment
in the eastern Mediterranean and its bordering continental areas.
The values of $\delta^2$H$_{fi}$ and $\delta^{18}$O$_{fi}$ fit within the range of values for modern precipitation, giving
confidence that these measurements do reflect past precipitation composition despite the influence of
multiple inclusion populations. The lack of apparent scatter towards positive $\delta^{18}$O values both in the
precipitation and fluid inclusion datasets further indicates that the data represent little-altered
precipitation values, and that surface re-evaporation was minor at least during humid phases.
However, the range of fluid inclusion values is inconsistent with either an exclusively Tunis-type or
an exclusively Bet Dagan-type moisture source for precipitation in Cyrenaica during MIS 3. Even
when all but the subset of fluid inclusion analyses who replicates are similar are excluded (Fig. 6), the
population is split between the Tunisian and Israeli precipitation end-members.

### Strontium isotopes

The $^{87}$Sr/$^{86}$Sr signal in the SC-06-01 record is rather invariable (Fig. 7), with all analyses indicating
values within analytical error. Mean values vary between 0.708275 and 0.708524 and although there
is an apparent trend from maxima at 34 and 64 ka BP with a minimum at 52 ka BP, which mimics the
precession history, this is too weak to be significant relative to the error.

### Calcite carbon isotopes

Both $\delta^{13}$C$_{cc}$ and $\delta^{18}$O$_{cc}$ show similar trends throughout the record (Fig. 8), indicating that depleted
oxygen isotopes coincide with depleted carbon isotope values. This does not appear to arise from



fractionation on the speleothem surface (Hoffmann et al., 2016), and so represents changes in soil
bioproductivity acting in concert with changes in precipitation.

## Discussion

### Moisture advection during Libyan humid phases

The range of values of both individual and replicated fluid inclusion measurements can only be
reconciled with multiple moisture sources. Most of the fluid inclusion data cluster between the
weighted mean value for precipitation collected at Sfax with a mixed source from the Atlantic and
western Mediterranean, ("Sfax Mixed" $\delta^{18}O_{ppt}$ = -4.93 ‰, $\delta^2H_{ppt}$ = -26 ‰; Fig. 9) and High
Precipitation events at Bet Dagan ($\delta^{18}O_{ppt}$ = -6.33 ‰, $\delta^2H_{ppt}$ = -21.46 ‰; Fig. 9). This value is
representative of many of the largest individual precipitation events at Sfax in the period 1992-1999
associated with a Western Mediterranean moisture source (Celle-Jeanton et al., 2001). However, the
fluid inclusion data cluster also extends to the end member reflecting pure western Mediterranean
sources at Sfax ($\delta^{18}O_{ppt}$ = -3.99 ‰, $\delta^2H_{ppt}$ = -20.3 ‰; Fig. 9), indicating a third end member
composition with higher $\delta^{18}O_{ppt}$. Consequently, we consider that this data reflects a dynamic balance
of moisture sources contributing to rainfall in Cyrenaica which resembles modern precipitation in
Tunisia and Israel in roughly equal proportions.
The weighted mean value for Atlantic-sourced precipitation events in Sfax ($\delta^{18}O_{ppt}$ = -6.7 ‰, $\delta^2H_{ppt}$ =
-37.7 ‰) is distant from any observed fluid inclusion value (Fig. 9). Likewise, compositions similar to
the high amount Atlantic-sourced rainfall events in Sfax ($\delta^{18}O_{ppt}$ = -8 ‰, $\delta^2H_{ppt}$ = -46 ‰) are not
reflected in the fluid inclusion data in Figure 9 suggesting a relatively low admixture of water from
this source. A simple 3-end-member unmixing of fluid inclusion isotope values using the quantitative
approach of (Rogerson et al., 2011) indicates that Atlantic-sourced water supplied no more than 15 %
of the mass for any given fluid inclusion analysis. However, the coherence of fluid inclusion isotope
ratios with the weighted mean of "mixed" Atlantic and Mediterranean precipitation at Sfax suggests
that this small Atlantic influence is nevertheless persistent, and this must reflect synoptic westerly
storms (Celle-Jeanton et al., 2001). An alternative way to explain the trend of some points towards





enriched $\delta^{18}O$ values on the GMWL would be the temperature-dependent fractionation that would be
caused by a shift to summertime precipitation. We do not favour this explanation as it requires a more
fundamental reorganisation of regional atmospheric circulation than our suggestion that the winter
storms observed today penetrated further east in the past.
Within the data presented in Figure 9, the Phase II fluid inclusions are exceptional, because none
show compositions consistent with a Bet Dagan source. Indeed, all the measurements for this period
resemble GMWL compositions. This seems to reflect a fundamental difference between this period
and Phases I and III, where all precipitation is drawn from synoptic westerly storms in the winter.
Consequently, it would seem that during the Obliquity-forced period of humidity the Israeli-mode
precipitation did not occur in the manner that it did during both Precession-forced periods of
humidity.
Although the isotopic composition of Mediterranean water will have been more enriched during MIS
3 due to ice-volume effects and increased Mediterranean water residence time (Rohling and Bryden,
1994), the similar mean values of the SC-06-01 fluid inclusion waters compared to modern
precipitation indicates the meteoric waterline at this time was not displaced to more enriched isotope
values. This could reflect balancing of source water effects by changes in kinetic fractionation during
evaporation (Goldsmith et al., 2017), which is controlled by normalised relative humidity. This would
imply that the Mediterranean air masses were less saturated with moisture than today during MIS 3,
which is consistent with the high deuterium excess $\delta^2H_{excess}$ values found in some fluid inclusion
samples (Fig. 10), but is difficult to reconcile with the increased precipitation recorded in SC-06-01.
Alternatively, the source water effect may be countered by increased runoff from the margins of the
Mediterranean supplying isotopically depleted water to evaporating surface water. Isotopic
"residuals" consistent with this argument are identified throughout MIS 3 in the eastern
Mediterranean marine core LC21 (Grant et al., 2016), and this is also consistent with higher rainfall in
Cyrenaica. We therefore favour the latter explanation.
We conclude that most of the precipitation supplied to Cyrenaica during MIS 3 was sourced from
within the Mediterranean basin, which exhibited a similar meteoric water cycle to that observed today



albeit with more freshwater influence. This is a critical observation, as internally-cycled water cannot
alter the basin-scale hydrological balance and therefore is a minor influence on deep convection in the
Mediterranean Sea (Bethoux and Gentili, 1999). The precipitation feeding runoff must be externally
sourced if it is to materially change Mediterranean functioning, as is observed during sapropel events
(Rohling et al., 2015). As most of the precipitation identified in SC-06-01 is sourced internally to the
Mediterranean, only the small, Atlantic-sourced portion of this water can be assumed to play a role in
Mediterranean freshening. This conclusion is likely transferable to any site on the continental margins
of the Mediterranean. This observation is critical, as it decouples the processes of precipitation on the
Mediterranean margins with sapropel formation, and consequent changes in moemtum transfer to the
North Atlantic (Rogerson et al., 2012). Consequently, we recommend that great care is taken to
determine whether past precipitation peaks reflect significantly enhanced external water advection
before any continental record can be used as a basis for inferring Mediterranean freshening.
**Palaeoclimatological significance**
The consistency of MIS 3 and modern precipitation isotope values permits comparison of fluid
inclusion values and precipitation magnitude records at Sfax and Bet Dagan. Most of the water
reaching Susah Cave seems to have been derived from large-magnitude rainfall sourced from the
Western or Eastern Mediterranean surface water. The primary difference between these end-members
is the level of $D_{excess}$, with the Western water ~10 ‰ and Eastern water ~30 ‰. This difference allows
the influence of these two sources to be compared between the three major humid phases (Hoffmann
et al., 2016) recorded in SC-06-01 (Fig. 10). These phases reflect changes in the distribution of
insolation as a consequence of changes in orbital tilt, with Phase I (65 to 61 ka BP) and Phase III
(37.5 to 33.5 ka BP) associated with reflecting Northern Hemisphere heating during precession
minima and Phase II (52.5 to 50.5 ka BP) which has been associated with a change in obliquity. In all
cases, the peak in rainfall recorded by the speleothem leads the orbital peak by ~3 ka. Phases I and III
both show very elevated $D_{excess}$, whereas no such values were found in Phase II. This provides further
support to our conclusion that the Eastern Mediterranean source contributed significant moisture to
Cyrenaica during precession-related humid events, but that it did not during the obliquity-related



humid event. This difference in the origin of the moisture feeding rainfall may explain the difference
in average $\delta^{18}O_{cc}$ during these different phases (Hoffmann et al., 2016).
The varying balance between Eastern and Western precipitation is diagnostic of changing basin-scale
atmospheric structure during the past. Eastern-sourced rainfall may occasionally relate to wintertime
storms, as today (Gat et al., 2003), but essentially reflects convective rainfall with relatively small
advection distances. The significant enhancement of the magnitude and regional significance of this
convective rainfall observed at Susah Cave must reflect greater atmospheric convergence due to
northward displacement of the annual average position of the ITCZ (Tuenter et al., 2003). Contrary to
this, the Western-sourced moisture is transported ~1500 km eastwards to reach Cyrenaica, which must
reflect the mid-latitude storm track (Brayshaw et al., 2009). Consequently, although it does not seem
that Atlantic moisture is important to the climatology of Cyrenaica, the momentum derived from
Atlantic winter storms predicted by regional climate modelling (Brayshaw et al., 2009) and observed
on the northern Mediterranean margin (Toucanne et al., 2015) remains pivotal to supplying moisture
to North Africa. Within obliquity-forced phases, advective transport of moisture alone drives
humidity. In contrast, we conclude that during precession-forced humid phases, the impact of
advective transport of moisture from the Western to the Eastern Mediterranean basin occurs alongside
strong convergence and convective rainfall within the eastern basin. The dilution of the advective
signal by internal convective rainfall may be the reason why Dansgaard-Oeschger cycles in the North
Atlantic are well reflected at Susah Cave during high precession (Hoffmann et al., 2016), whereas
there is weaker correspondence of Cyrenaican rain and North Atlantic heat during low precession.
Further constraint on large-scale atmospheric advection can be provided by Sr-isotopes, which are
known to be sensitive to changes in transport of Saharan dust (Frumkin and Stein, 2004). Even
considering the most slow-growing and most rapidly-growing parts of SC-06-01, no significant
difference in $^{87}Sr/^{86}Sr$ was identified. This is unexpected and significant, as climate-driven changes in
$^{87}Sr/^{86}Sr$ have previously been reported from speleothems in the Mediterranean region (Frumkin and
Stein, 2004). It seems that despite changes in the intensity of moisture transport during the period 65-
30 ka BP, there is no large-scale change in atmospheric dust transport direction. This further supports



our conclusion from the fluid inclusions that the Eastern Mediterranean rainfall operating during
precession minima reflects enhanced internal convection rather than transport of moisture from the
east or south with an atmospheric circulation pattern that prevails today.
*Implications for Susah Cave $\delta^{18}O_{cc}$*
Aside from those data with high deuterium excess, which reflect influence from the Eastern
Mediterranean source, much of the variance in the fluid inclusion dataset is captured by a two end-
member mixing system resembling modern rainfall in Tunisia. One end-member is the Western
Mediterranean source of Celle-Jeanton et al (2003), but the other is isotopically too heavy to be
identified with the Atlantic source. Rather, it resembles the "Sfax Mixed" population defined by
Celle-Jeanton et al (2003), reflecting a mixed source of moisture from both the Western
Mediterranean and Atlantic. Consequently, although quantitatively minor amounts of Atlantic water
reached the site, changes in the moisture advection driven by westerly winds had a strong influence on
$\delta^{18}O_{dripwater}$ trends in time. At Sfax today, this influence causes a prominent bimodal behaviour with
two rainfall maxima with different $\delta^{18}O_{ppt}$, which eliminates a simple and quantitative rainfall amount
control on precipitation, which can be observed at Tunis (WMO code 6071500,
https://nucleus.iaea.org/wiser/gnip.php). Furthermore, addition of heavy rain events derived from the
Eastern Mediterranean aliases the tendency towards depleted $\delta^{18}O_{dripwater}$, as this water is also more
depleted than modern Western Mediterranean precipitation. In the Bet Dagan data, there is also a
tendency to lower $\delta^{18}O_{ppt}$ with higher precipitation amount, but the relationship between rainfall
amount and rainfall isotope composition is not identical to Tunis. Ultimately, it seems likely that
rainfall amount changes at Susah Cave do cause depleted (enriched) $\delta^{18}O_{cc}$ values to be associated
with high (low) rainfall, but this is too complicated by independent changes increases (decreases) in
westerly moisture advection and increases (decreases) in convergence. Qualitatively, all these
parameters are expected symptoms of North African humid phases and so these trends remain a
valuable expression of climatic variability. Quantitatively, more information is required to translate
the trends into fully-functional palaeoclimatologies, and this analysis pivots on whether $\delta^{18}O_{cc}$ trends
reflect changes in water deficit / surplus in Cyrenaica.



Although it is likely the oxygen isotope fractionation during calcite precipitation occurred close to
isotope equilibrium (Hoffmann et al., 2016), there is a good degree of correspondence between
positive and negative phases in $\delta^{18}O_{cc}$ and $\delta^{13}C_{cc}$, indicating a shared control. Indeed, $\delta^{13}C_{cc}$ has a
markedly higher amplitude variability than $\delta^{18}O_{cc}$. More isotopically depleted carbon may represent
increased incorporation of respired soil carbon, increased dominance of C3 over C4 plants, and/or
decreased degassing of aquifer water (Baker et al., 1997). Today, the Susah Cave location on Jebel
Malh has very thin soil cover, colonised by shrubby maquis vegetation. Soil respiration and
colonisation by C3 plants is limited by the strong water deficit of the region, and aquifer water
outgassing is enhanced by long residence times due to low water infiltration. Increased water
availability will progressively deplete the $\delta^{13}C$ of dripwater by all three mechanisms described above.
Consequently, all three of these processes promote correlation between $\delta^{13}C_{cc}$ and precipitation
amount. Within the $\delta^{18}O_{cc}$ data series, peak growth rates occur both during relatively enriched and
relatively depleted isotope stages. This is not the case for $\delta^{13}C_{cc}$, which more consistently shows
depleted values during times of rapid growth (SC-06-01 growth phases shown in Fig. 11). We
therefore consider it likely that $\delta^{13}C_{cc}$ indeed more accurately records rainfall amount than $\delta^{18}O_{cc}$ does.
Conclusions and Implications
A key feature of this combined dataset is the long-term sinusoidal trend in both the $\delta^{18}O_{cc}$ and $\delta^{2}H_{fi}$,
reflecting the differing rainfall regimes dominant between Humid Phases I and III compared to Phase
II. This is not developed in $\delta^{13}C_{cc}$, implying that the process forcing the long-term cycle in moisture
source is not impacting on carbon dynamics in the soil and epikarst. We therefore conclude that there
is a mixed amount and source control on $\delta^{18}O$ and $\delta^{2}H$ in the SC-01-06 record, whereas $\delta^{13}C$ is
dominantly controlled by water availability.
The fluid inclusions from SC-06-01 show that rainfall compositions in the southeast Mediterranean
region during MIS 3 were comparable to modern rainfall compositions recorded in regional GNIP
datasets. However, the diversity of compositions is impossible to explain with a single rainfall source,
rather indicating that moisture derived from the Atlantic, the Western Mediterranean and the Eastern



Mediterranean basins have all contributed to MIS 3 precipitation in Libya. This requires both
enhanced westerly advection of moisture to this region, reflecting the Atlantic storm track, and
enhanced convective rainfall within the Eastern Mediterranean basin. There is some indication that
these two mechanisms differ in terms of their response to orbital forcing, with precession minima
enhancing westerly advection and internal convection, whereas obliquity minima enhance westerly
advection without significantly altering internal convection.
Crucially, this picture is most consistent with atmospheric circulation over the Mediterranean
remaining essentially unchanged during precession cycles. This is consistent with regional climate
model experiments showing major enhancement of winter westerly storm activity, but it not
consistent with the extreme migration of the ITCZ, where the monsoon belt approaches the North
African coast. The strong implication is that a significant arid belt is retained between the
Mediterranean and the ITCZ, even when northernmost Africa is experiencing significantly enhanced
rainfall.
It is likely that rainfall amount played a role in controlling the isotopic composition of the calcite in
this speleothem ($\delta^{18}O_{cc}$). However, the more depleted values reflecting higher rainfall are also
consistent with different mixing between the end members identified by the fluid inclusion analysis.
The structure of the $\delta^{13}C_{cc}$ record provides an independent means of assessing changes in water
surplus / deficit, as more depleted values will reflect lower aquifer residence times, enhanced soil
respiration and changes in vegetation structure, all of which are limited by water availability in this
semi-arid environment. Combined analysis of the proxies provides a powerful new demonstration that
the northeast Libyan climate was more humid during millennial-scale warm periods in the North
Atlantic realm, but quantification will be dependent on generating unambiguous independent evidence
for water availability in the soil and epikarst.
Acknowledgements
We thank the Royal Geographical Society for the pump-priming investment that began this work
(Thesiger-Oman International Fellowship 2009), the Natural Environment Research Council for





providing the funds that made the analytical work on this project possible (NE/J014133/1) and The
Leverhulme Trust for funding activities within the associated International Network (IN-2012-113).

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

Figure Captions
Figure 1: Map showing the location of Susah Cave (filled circle) and GNIP sites used in the discussion
(open circles).
Figure 2) Macroscopic structure of SC-06-01 speleothem, showing alternation of transparent and
milky fabrics
Figure 3) Variability of water content (μL) per unit mass of speleothem (g) in SC-06-01 fluid inclusion
samples. Grey area shows working range of instrument.



Figure 4a) Fluid inclusion oxygen isotope values ($\delta^{18}O_{fi}$; blue crosses) compared to calcite oxygen
isotope values ($\delta^{18}O_{cc}$; black circles and line); 4b) Fluid inclusion hydrogen isotope values ($\delta^2H_{fi}$; blue
crosses) compared to $\delta^{18}O_{cc}$ (black circles and line). Growth Phases I, II and III are shown as grey
areas.
Figure 5a) Regional precipitation isotope data. Thick line represents Global Meteoric Water Line,
dashed thick line represents Mediterranean Meteoric Water Line and thin lines representing
expected range of deviation (±10 ‰ $\delta^2H_{ppt}$) below GMWL and above MMWL. Bet Dagan, Tunis and
Sfax GNIP datasets (http://www-naweb.iaea.org/napc/ih/IHS_resources_gnip.html). Sfax Atlantic
and Mediterranean Rainfall are taken from Celle-Jeanton et al. (2003). 5b) Summarised precipitation
isotopes, and fluid inclusion measurements for SC-06-01.
Figure 6) Double-replicated fluid inclusion measurements from SC-06-01, and regional precipitation
isotope trends.
Figure 7) $^{87}Sr/^{86}Sr$ record for SC-06-01, compared to calcite $\delta^{18}O_{cc}$ record (light grey line). Error bars
are 2σ. Growth Phases I, II and III are shown as grey areas.
Figure 8) Carbon isotope ($\delta^{13}C_{cc}$) record for SC-06-01 compared to oxygen isotope record ($\delta^{18}O_{cc}$;
(Hoffmann et al., 2016)). Growth Phases I, II and III are shown as grey areas.
Figure 9) Fluid inclusion measurements relative to summarised precipitation data and the modern
precipitation end members used in the discussion. Solid lines are the Meteoric Water Lines as in Fig.
5a. Precipitation and fluid inclusion measurements are as shown in Figure 5b. "Mean Atlantic", "Sfax
Mixed", "Sfax Med" and "High Precip Atlantic" indicate the mean of measurements in Celle-Jeanton
et al (2003) originating from Atlantic moisture, mixed source, Mediterranean moisture and High
Precipitation measurements from an Atlantic moisture source (as described in Discussion)
respectively. "Mean Bet Dagan" is the mean of GNIP measurements from this location, and "High
Precip Bet Dagan" is the subset of high precipitation measurements as described in the Discussion.
Figure 10) Fluid inclusion deuterium excess ($\delta^2H_{excess-FI}$) relative to calcite $\delta^{18}O_{cc}$. Note some fluid
inclusions (70 to 60 ka BP and 40 to 30 ka BP) show high $D_{excess-FI}$ indicative of an Eastern
Mediterranean source. Growth Phases I, II and III are shown as grey areas.





**Figure 2**

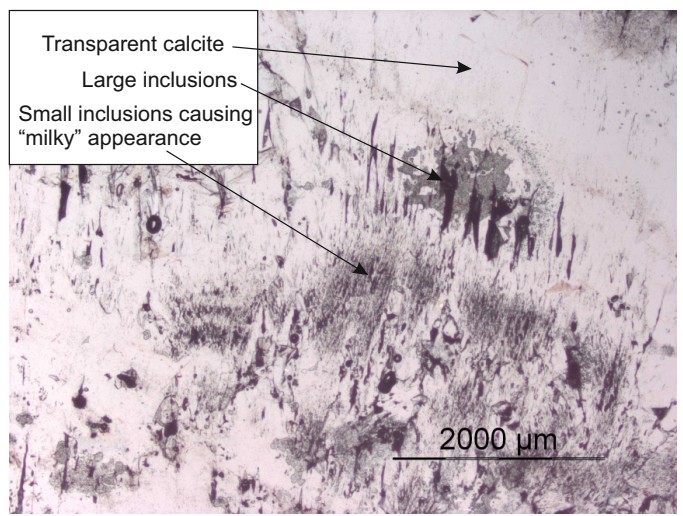



## Figure 3

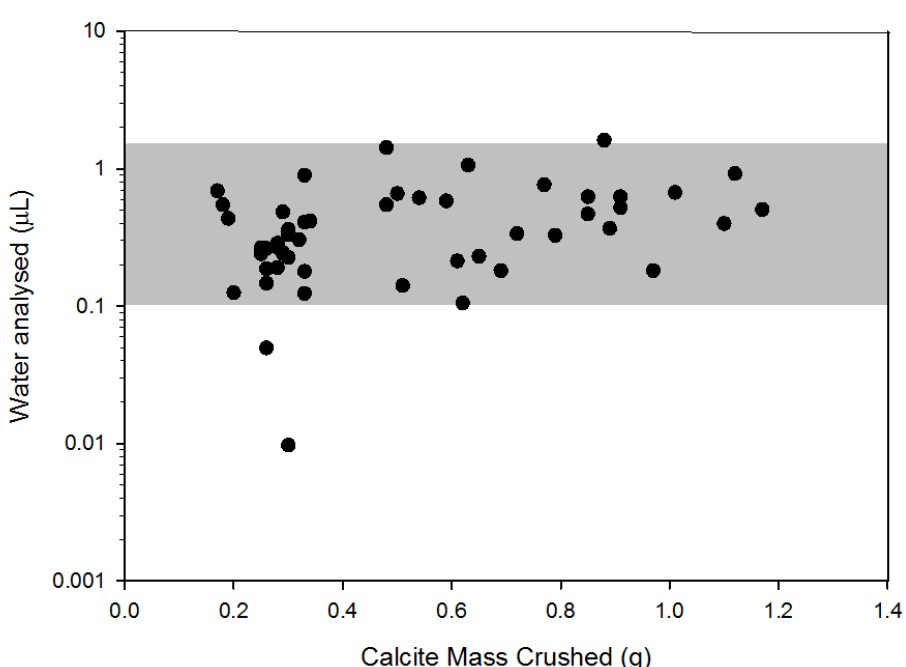





Figure 4a

Figure 4b




## Figure 5a

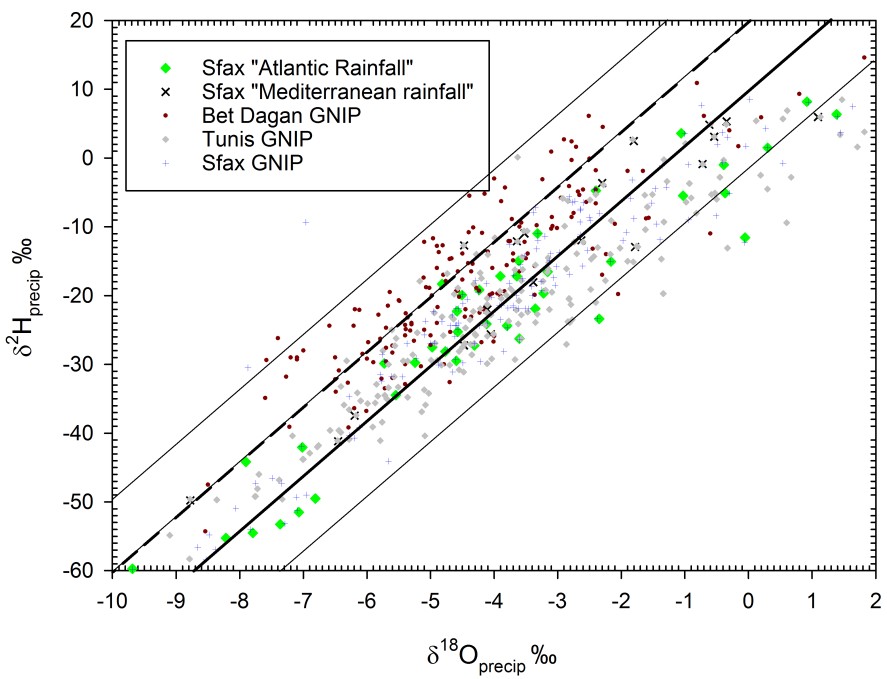

## Figure 5b

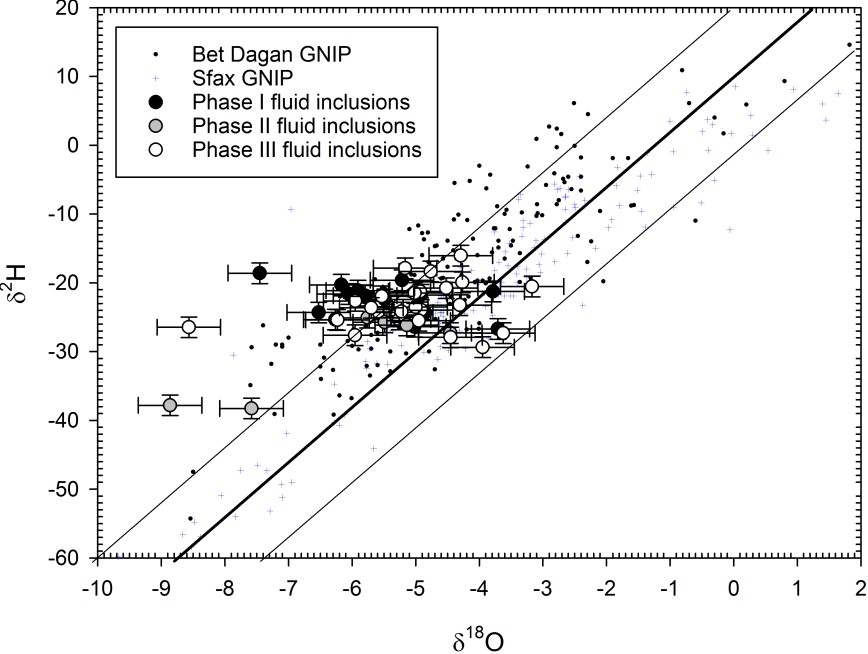

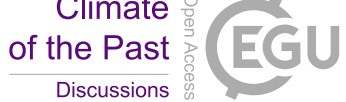



## Figure 6

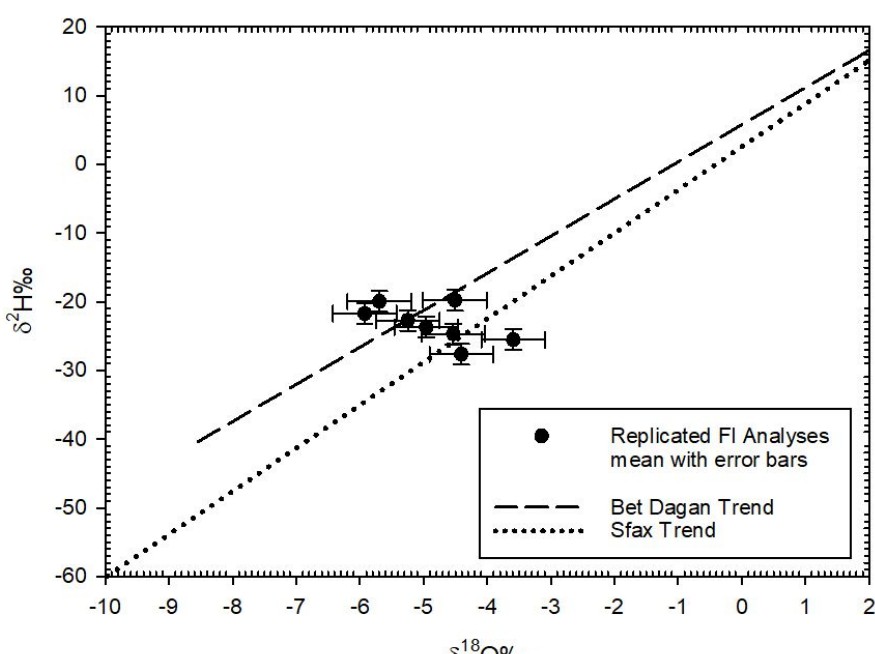





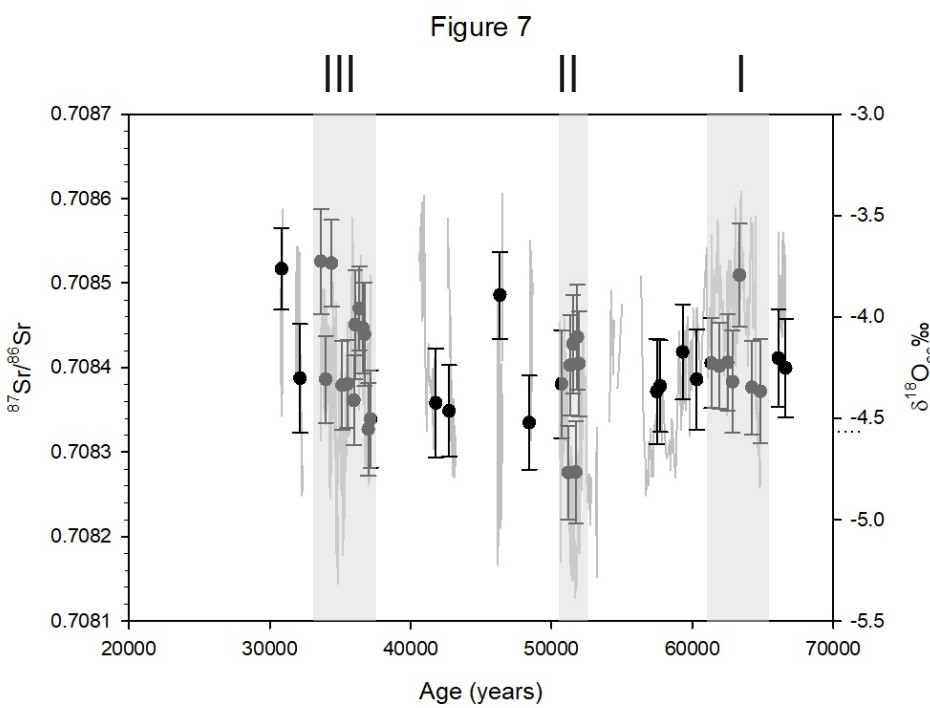



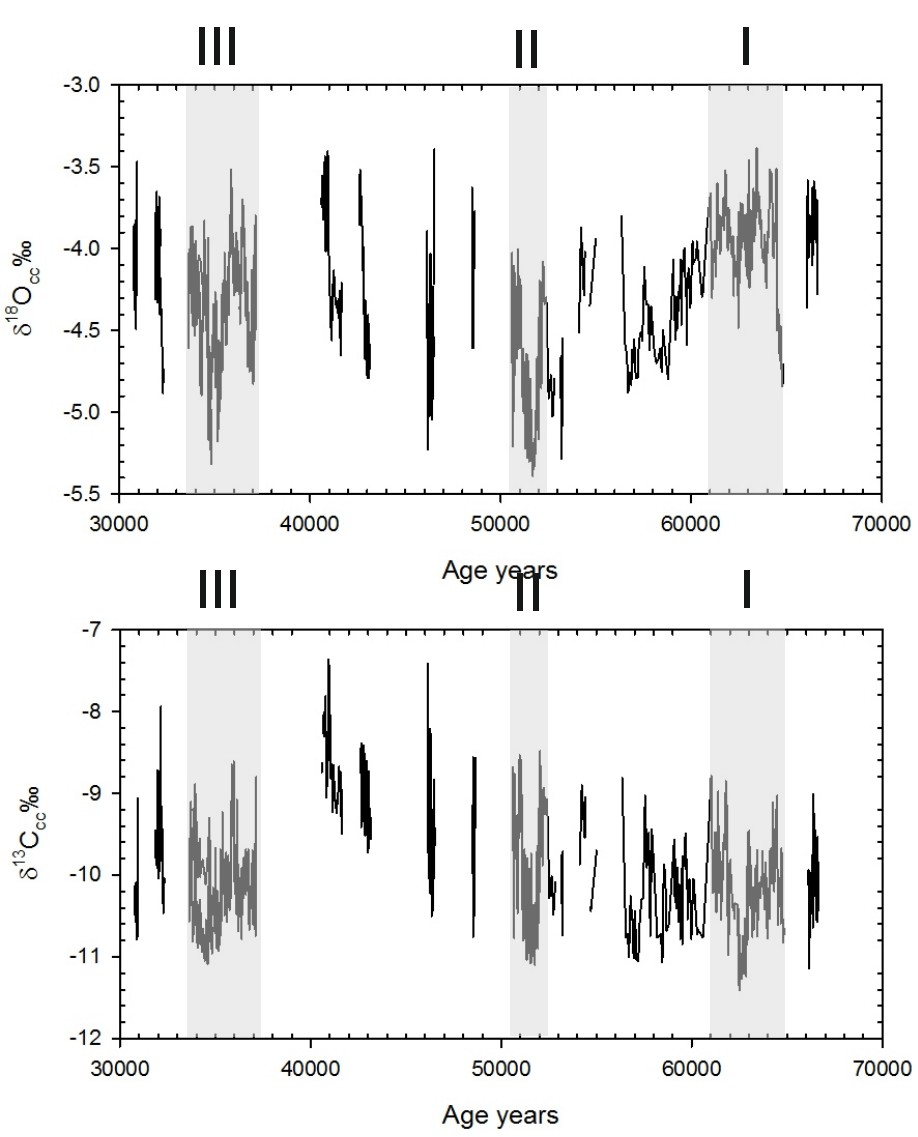





## Figure 9

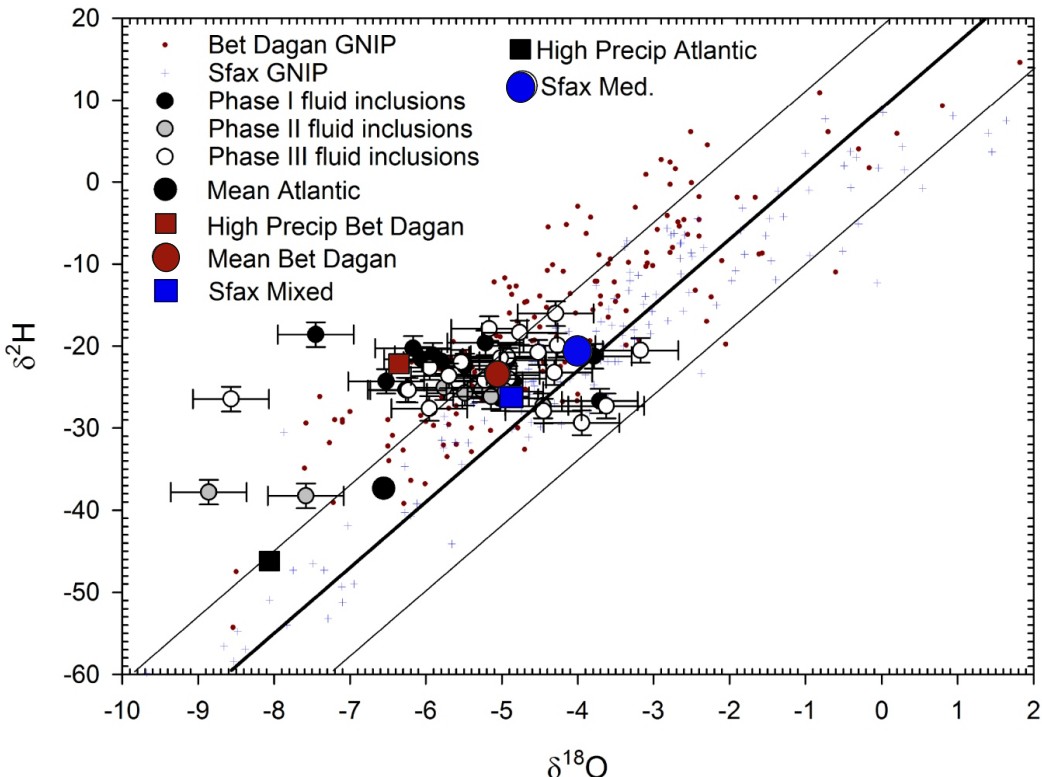





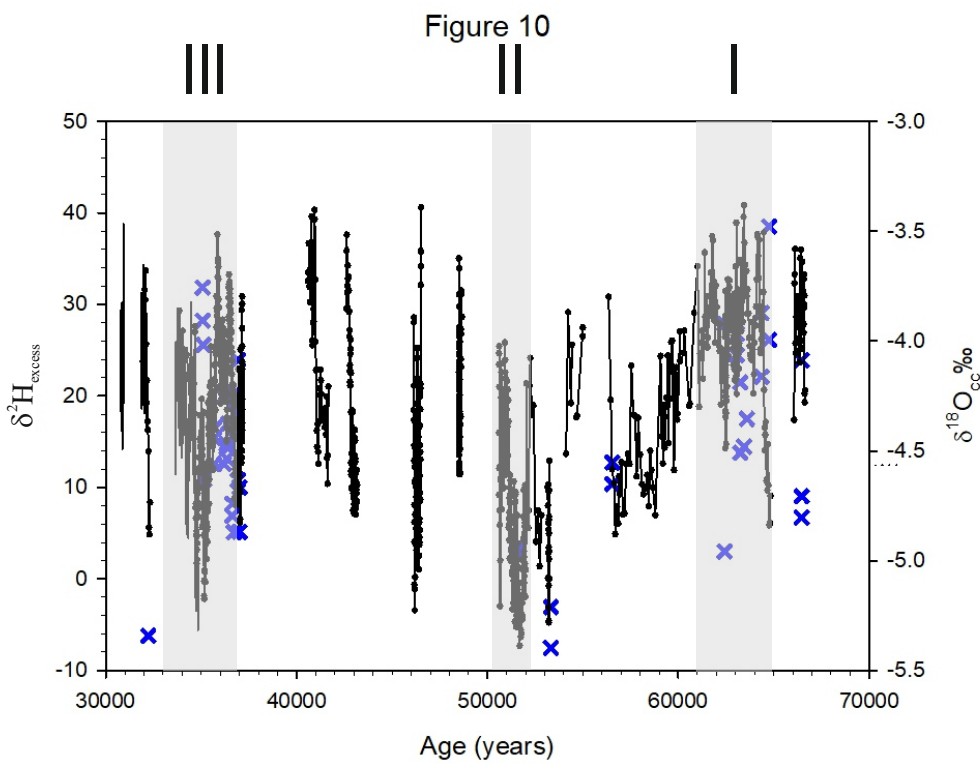

Figure 10