# Peer review of "Enhanced Mediterranean water cycle explains increased humidity during MIS 3 in North Africa"

_Climate of the Past, 2018_

## Referee Comment (RC1) · Anonymous Referee #1 · 2 Dec 2018

**General comments**

This manuscript presents new analyses of the stable isotopic composition of fluid inclusions in a speleothem from Susah Cave, Lybia. The stable isotopic compositions and dating of this speleothem had previously been published revealing three major periods of speleothem deposition (Phase I- III) covering the time intervals between 65-61 ka, 52.5-50.5 ka and 37.5-33 ka, respectively. Phases I and III coincide with periods of low precession parameters (high northern hemisphere summer insolation) whereas Phase II suggests increased moisture availability in a phase of high obliquity. The comparison of the fluid inclusion stable isotopic composition and d-excess values with stable isotopic compositions of present-day rainfall from the region and the stable isotopic values of speleothem carbonate suggest that the rainfall sources were different in

the precession triggered humid phases (I and III) than in the obliquity triggered phase (II). Consistent strontium isotope compositions across all three phases indicate that there was no major shift in Sahara dust transport to Susah Cave. The shift of moisture sources between the main deposition phases is therefore attributed to a change in the contribution from mainly westerly rainfall systems in Phase II to a mix of westerly and convective eastern sources in Phases I and III.

This is the first study of fluid inclusions in speleothems in northern Africa and the research presented is both within the scope of Climate of the Past and relevant. The scientific methods are valid and well-described, however clarity manuscript could be improved by changing the structure especially in the Discussion section. The data presented here lead to some substantial and new hypotheses about rainfall moisture sources during wet periods in the northern African coastal regions, however the current structure of the manuscript makes it hard for the reader to follow the arguments and there are a few important points concerning the precesses leading to humid periods in northern Africa and the factors influencing the stable isotopic composition of rainfall that need to be clarified/included. Proper credit is given to related work, however in some places in the manuscript the distinction between analyses done for this study and analyses of Hoffmann et al., (2016) could be made more clear (see Technical corrections).

The formation of north African pluvials and sapropels and the time period covered by the samples: The section 'Past climate changes in North African hydroclimate' in the Introduction gives a good overview of general climatic variations across North Africa in the past. However I disagree with some of the statements made in the paragraph starting in line 66. The authors state that records of past lake levels and vegetation (pollen) suggest wet conditions across the Sahara an that this wetting was caused by a northward shift (up to 30-35N) of tropical rainfall associated with the ITCZ, they then continue to counter this interpretation with model studies that have shown that tropical rainfall only extended as far as $\sim$25N. I don't see this contradiction as in my

understanding of the causes of North African pluvials tropical rainfall only extended far enough north to reach the southern edges of central and northern Saharan drainage basins, providing an initial source for moisture that was then recycled within the region (e.g. Rohling et al., 2004, 2002). A northward shift of the tropical monsoon belt to ∼25N would be enough for this. I would also like to see a bit more detail about what the lake and vegetation records from the Sahara suggest for the actual time period covered by the speleothems. Most of the statements in the overview of past climates in the region are very general and refer to sapropel events, however the speleothems do not cover any periods of actual sapropels. In fact, most sapropels occur during interglacial phases and the speleothems here formed during a glacial time. A short discussion of the effects of the different boundary conditions and how it could affect northern African moisture should be included. I also think there should be a section in the introduction about the present-day rainfall systems in the region and at the rainfall sampling stations that are used in the discussion. This section should also summarize the publication by Celle-Jeanton et al. (2001) and explain how the different rainfall end members were defined (the publication is cited in text as Celle-Jeanton et al., 2003 but there is only a publication from 2001 in the reference list, which one is correct?). What are the characteristics of the end members? How were they defined – e.g. GNIP data are usually sampled at monthly intervals, how were high rainfall events at Bet Dagan separated from that and how much rainfall is high rainfall? Do the end-members have different local meteoric water lines and d-excess? How were the different moisture sources assigned to monthly rainfall - Back trajectories, analyses of daily synoptic charts? What factors other than moisture source drive the variation of d18Oppt and d2Hppt at the rainfall stations? How were the averages shown in Figure 9 calculated – simple arithmetic mean or amount-weighted means? What synoptic processes were involved in the formation of rain clouds – convection, advection? What circumstances lead to convective rainfall in the region? Further adding to the point above, it seems that the discussion of factors influencing stable isotopic composition of rainfall is focused on only the effects of rainfall amounts, temperature and moisture source. These

are important factors, but I think one important factor is missing and that is the effect of cloud formation processes on the stable isotopic composition of rainfall. One conclusion of the manuscript is that during phases of high precession, rainfall at Susah cave originates from a combination of western Mediterranean and eastern Mediterranean sources with a small addition of Atlantic rain. Present-day rainfall at Bet Dagan is considered to represent the end-member composition of the eastern Mediterranean source. As a mechanism for the transport of moisture to Susah Cave from the east, strong regional convection is suggested. However present-day rainfall in Bet Dagan is dominated by mid-latitude winter storms. The cloud formation processes of these winter storms and convection will affect the stable isotopic composition of the resulting rainfall (Aggarwal et al., 2016) and this difference is not considered in the manuscript.

The discussion is structured in a very confusing way. At the moment the discussion of the d18Oppt and d2Hppt are in one chapter and only in the second discussion chapter we are told "The primary difference between these end-members is the level of Dexcess...". I think the arguments made for the mixing of the different end members would be much more clear if the discussion of the stable isotopic composition and the d-excess would be combined including clear definitions of the values for the three depositional phases of the speleothem and the modern rainfall end-members. One example for this is the sentence starting in line 249 which states that Phase II fluid inclusions are inconsistent with Bet Dagan rainfall, however in Figure 9 (which is referenced in the sentence) most of the values plot right next to the Bet Dagan mean values. It is not clearly stated here why the Bet Dagan mean cannot represent the same rainfall source as Susah cave and including the d-excess values right away would solve this problem.

There seems to be a discrepancy between some of the statements in the manuscript with regards to the Atlantic rainfall source. It is stated that a small Atlantic source of rainfall can be assumed for Susah cave and that this conclusion is likely transferrable to any site on the continental margins of the Mediterranean (Lines 276-279). On the other hand, moisture sourced from the Atlantic is also suggested as the only possible source

for a freshening of the surface water of the Mediterranean that counteracts the ice volume effect. I think this needs to be clarified. This should also include an argument why runoff from northern Africa originating from recycled monsoon rainfall is not an option. Another contradiction is between the statements in lines 304 -306 and 325-328. First it is stated that increased convection during phases with a low precession parameter must be related to a northward shift of the ITCZ, then the convection is attributed to enhance internal convection.

Specific comments by chapter

Introduction - The central North African speleothem record

2. The last paragraph of the section starting in line 123 is not really about 'the central North African speleothem record' but rather an introduction to fluid inclusions and should maybe be in its own chapter.

Material and methods

The speleothems carbonate stable isotopic composition were published by Hoffmann et al., (2016), so the sentence about their measurement in line 145 can be removed. Results Fluid inclusions

I think this section should include a clear defienition of the three depositional phases, giving the range of values of fluid inclusion stable isotopes and d-excess. This would make the comparison of the much easier.

Calcite carbon isotopes

As mentioned before, these were included in the Supplement of the publication by Hoffmann et al. (2016) and are not results of this study. I think this section can be cut.

Technical corrections

I have a few technical comments about the text: The abbreviation used for the d-excess parameter is not consistent throughout the manuscript Similarly, the stable isotopic

composition of rainfall is called $\delta$18Oppt and $\delta$2Hppt in the text, but in Figure 5a the axes labels are $\delta$18Oprecip and $\delta$2Hprecip. Throughout the manuscript the authors talk about precession maxima and minima – precession is a directional value and does not have high or low values, what is referred to here is the precession parameter which has maxima and minima. I saw in several places that there was no '.' after et al. For future submissions I would recommend changing the formatting of the reference list to help reviewers find the references they are looking for I would also add some the title of the publication and author list a the top of the supplementary information.

Figures

I think that the Figures could be improved a lot, some of them are not easy to read and a bit confusing.

Figure 1: I think adding some more information to the map would be useful – e.g. where are the different moisture sources that are considered end members contributing to Susah rainfall and what does the present-day atmospheric circulation look like?

Figure 4: The actual data presented by this publication is plotted behind the carbonate stable isotope data in this figure and it looks like the shading indicating the three main deposition phases is also overlaid onto the data presented. This makes it really hard to see the actual data being presented. Same applies to Figure 10.

Figure 5a: I find this Figure hard to read. It looks like the Sfax "Atlantic rainfall", Sfax "Mediterranean rainfall", Tunis GNIP and Sfax GNIP are all very similar. The small crosses used for the Sfax GNIP data are unlikely to be visible once this is scaled down for the pdf version of the publication (this also applies to Figure 5b and 9) Figure 5b: I find it almost impossible to distinguish between the different speleothem deposition phases in this Figure. I think splitting it into three subfigures would make sense. In each subfigure I would still include all the data but plot only the data for one of the Phases with black error bars and black borders of the dots while the other two are shown in grey. I would also make sure that the Phase that is being highlighted is plotted on top

of everything else. Similar problems apply to Figure 9 – I think Figure 9 might actually be cut and the few points shown there could be included here.

Figure 6: would it not make sense to show which of these samples are from which depositional phase?

Figure 7: The actual data presented are much more visible here than in Figure 4 and 10, but there are no data between 20000 and 30000 years and that part could be cut, also here the shading for the boxes marking the main growth phases could be plotted behind the data rather than overlaid on them.

Figure 8: The two subfigures are over one another here, the second panel should be moved down a bit. It looks like the d13C record is duplicated between ∼33000 and 36000 years. I don't think this is explained in the paper.

References:

Aggarwal, P.K., Romatschke, U., Araguás-Araguás, L.J., Belachew, D., Longstaffe, F.J., Berg, P., Schumacher, C., Funk, A., 2016. Proportions of convective and stratiform precipitation revealed in water isotope ratios. Nat. Geosci. 9, 624–629. doi:10.1038/ngeo2739 Celle-Jeanton, H., Zouari, K., Travi, Y., Daoud, A., 2001. Caractérisation isotopique des pluies en Tunisie. Essai de typologie dans la région de Sfax. Comptes Rendus l'Académie des Sci. - Ser. IIA - Earth Planet. Sci. 333, 625–631. doi:10.1016/S1251-8050(01)01671-8 Hoffmann, D.L., Rogerson, M., Spötl, C., Luetscher, M., Vance, D., Osborne, A.H., Fello, Nuri, M., Moseley, G.E., 2016. Timing and causes of North African wet phases during the last glacial period and implications for modern human migration. Nat. Sci. Reports 6, 36367. doi:10.1038/srep36367 Rohling, E.J., Cane, T.R., Cooke, S., Sprovieri, M., Bouloubassi, I., Emeis, K.C., Schiebel, R., Kroon, D., Jorissen, F.J., Lorre, A., Kemp, A.E.S., 2002. African monsoon variability during the previous interglacial maximum. Earth Planet. Sci. Lett. 202, 61–75. doi:10.1016/S0012-821X(02)00775-6 Rohling, E.J., Sprovieri, M., Cane, T., Casford, J.S.L., Cooke, S., Bouloubassi, I., Emeis, K.C., Schiebel, R., Rogerson, M.,

Hayes, A., Jorissen, F.J., Kroon, D., 2004. Reconstructing past planktic foraminiferal habitats using stable isotope data: a case history for Mediterranean sapropel S5. Mar. Micropaleontol. 50, 89–123. doi:10.1016/S0377-8398(03)00068-9
* * *

---

## Referee Comment (RC2) · Anonymous Referee #2 · 10 Dec 2018

General:

Rogerson and coworkers present an interesting and comparatively large set of speleothem fluid inclusion isotope data from North Africa (Libya). It concerns follow up work on a publication by Hoffmann et al. (2016) who already presented the speleothem calcite d18O and the age model of this stalagmite. The ms is well written, provides sufficient background, and existing work is properly referenced for as far as I can oversee. The new fluid inclusion isotope dataset, that forms the backbone of this study seems technically sound (although unfortunately not presented in a table), but, as the authors describe, is also somewhat problematic because of variable water yields during analysis and relatively poor correlation of fluid inclusion isotope data with existing speleothem calcite d18O data. In any case, there is reasonable agreement

with rainfall isotope data from the area, lending support to the general accuracy of the dataset. An additional set of Sr isotope values of the speleothem appears to suggest that dust source changes play a minor role over the studied time interval.

Coming back to the fluid inclusion dataset, it indeed is somewhat uncomfortable that there is such poor correlation between fluid inclusion d18O (d18Ofi) and calcite d18O (d18Occ) and that no clear stratigraphic trends are visible in d18Ofi. It may be that this reflects high frequency rainfall sourcing changes or amount effect, but then why is this variability not visible in the d18Occ (which presumably has the higher stratigraphic resolution)? What I would like to see in the ms is an assessment of the extent to which the fluid inclusion isotope data are in isotopic equilibrium with the calcite (which is the assumption underlying the expectation that d18Ofi and d18Occ should be coupled to some extent). While isotope equilibrium is not a given for many speleothems, at least some consistency is to be expected between d18Occ and d18Ofi. If there is none or very little, one should think about what the fluid inclusion isotope values really represent. Based on petrographic observations the authors rule out the possibility of diagenetic alteration affecting the oxygen isotope values of the inclusion water, but in doing so, seem to offer no alternative interpretation to explain the absence of isotope equilibrium between speleothem carbonate and inclusion water. Maybe the observation that two phases of fluid inclusions are present in thin section study is of importance, although I can't directly see what kind of effect that would have on your analyses. It would perhaps be useful if the authors discuss that in a bit more detail in the context of their interpretation of the record.

A strong point of this study is that the authors duplicated fluid inclusion isotope data, and show the "duplicable" set in Fig 6. This set is significantly smaller than the original set, attesting to the analytical difficulties the authors describe in the earlier sections of the ms. Some potential flyers in the FI data in fig 4 do not duplicate, suggesting that these are indeed inaccurate data points, which in turn underlines the importance of this reproducibility test. The duplicable set spans a comparatively small range in

isotope values. Nevertheless, the case can be made that the dataset is influenced by two isotopically distinct atmospheric water sources. I would indeed agree that the FI isotope pattern fits a typical Mediterranean case of mixing between a source that, in deuterium-excess values, is closer to Eastern Mediterranean moisture and one that is closer to more Western sources. Bringing a third water source in, as is suggested in the ms, is not really supported by the data from my perspective. I generally doubt if one can simply take average isotope values of present day rainfall records, that have rather wide and overlapping ranges of isotope values, as distinctive fingerprint for past changes between these moisture sources. One could perhaps argue that slight isotope changes within each of these moisture sources can cause similar isotope patterns?

In summary:

I believe the dataset as presented should be interpreted primarily based on the duplicable dataset. This probably requires the authors to simplify their interpretation to some extent. I believe there still is an interesting story when one does so, as the unusual negative correlation between d2H and d18O in the dataset indeed is best explained by variable deuterium-excess values in the past, and therewith possibly with moisture source changes (presumably between the E and W Mediterranean basins) at the Susah site. Definitely more difficult to understand is the relatively poor correlation of d18Ofi values with corresponding d18Occ. Such a poor correlation practically implies that (near) isotopic equilibrium can't be demonstrated, and calls for caution in the interpretation of what the fluid inclusion isotope data (particularly the oxygen isotopes) really capture. Perhaps the same goes for the apparent absence of stratigraphic trends in the fluid inclusion isotope data as presented (with the possible exception of the d2Hfi data). So, while I am somewhat critical about some aspects of the interpretation of the data, I support the publication of this interesting study as it is one of the first examples that I have seen to demonstrate changes in deuterium excess in speleothem fluid inclusion isotope data, in an area where such changes can be quite large and have clear paleoclimatological significance.

Some further comments and questions:

I'd like to know where your duplicable samples from Fig 6 are located in the stalagmite (stratigraphically). All in one period, or distributed all over? Do you have a better correlation with the d18O values of the carbonate when you consider the duplicable dataset only? Further, I'd like to see if, based on the duplicable set only, one can still observe clear differences between the three wet intervals. If not, the interpretation of interval two as being different from one and three in its fluid inclusion isotope composition may be on thin ice. Not knowing where precisely the duplicable data fit in the stratigraphy, I can't assess this in the present ms.

Figures in general could be improved to make them more accessible for the reader.

Particularly figure 4a and b are not easy to decipher for me as fluid inclusion data points are partly hidden behind other graph elements. In period 2 in this figure, I note 4 d18Ofi and only 2 d2Hfi values (correct?). I assume the data are produced in pairs, so why are some d2H values missing? I'd like to see the data table for the fluid inclusions added to the ms, so that these data are accessible.

The term d2Hexcess (in Fig 10) is to my knowledge not the generally accepted notation for deuterium excess.

Line 191: GNIP does not stand for "Global Natural Isotopes in Precipitation", but for "Global Network of Isotopes in Precipitation".

It could be interesting to know a little bit more about the Sr isotope composition of the host limestone. Are your Sr isotope values merely representing the limestone host rock, or do you have reason to believe there is a Saharan dust siliciclastic component supplying some Sr to the epikarst as well? Could you have any sea spray effect? (not sure how far away from the sea you are)

Towards the end of the discussion, d13Ccc plays an important role. These data, however, are not shown (and they are not shown in any of the figures in Hoffmann et al 2016

either). It would be good to show the d13Ccc data if you claim that d13Ccc doesn't cycle like d18Occ does (line 371). To follow up on that; he entire argumentation in lines 369-374 is a bit puzzling to me because you don't mention your d18Ofi at all (apart from a "generic" d18O in line 373 with which you mean d18Occ I presume). Shouldn't d18Ofi do the same as your d2Hfi is your claim is correct?

Your statement in line 392 to 394 is not clear to me.

---

## Author Comment (AC1) · 29 Jan 2019

We thank the two anonymous Reviewers of our draft manuscript for their detailed and constructive reviews, and are extremely pleased that they find our work both interesting and worthy of publication. We fully concur that the interpretation of the data we present is complicated by structure of the dataset, and we are happy both reviewers agree with us that the data itself is so unique as to make a pressing case for publication and, that our analysis of it is fair, balanced and reasonable.

Below, we respond to the comments in order. The location code given for each comment and response represents the page the comment occurs in, followed by paragraph and lines (C, , -). Anonymous Reviewer 1

C2, 2, 3-4: ".....however, the clarity of the manuscript could be improved by changing the structure, especially the Discussion section." and C2, 2, 6-10: ".....the current structure of the manuscript makes it hard for the reader to follow the arguments.......... and the factors influencing the stable isotopic composition of rainfall that need to be clarified / included." Response: obviously, we set out to make the draft manuscript as clear as possible, but we are happy to clarify further through editing and re-structuring as recommended by this reviewer. This will include re-ordering the Discussion so that information arrives in the most useful order, and further improving the clarity of the figures. It is important to us to make our work as accessible as possible! The "few additional points" are discussed below.

C2, 3, 4-9 continuing to C3, 1, 1-5: "However, I disagree with some of the statements made in the paragraph starting in line 66.......... A northward shift of the tropical monsoon belt to 25oN would be sufficient for this." Response: The Reviewer's concern is essentially that we have exaggerated the lack of agreement between empirical evidence of wet conditions between 30-35oN and models, which generally do not get the monsoon so far north. They are also concerned we neglect the role of water recycling in the region 25-35oN. We are very happy to improve the discussion by including the water recycling argument, which we do overlook. On the other hand, we consider that the uncertainty in reconciling the empirical and physical lines of evidence is interesting and unresolved, and deserves to be highlighted in the way we do.

C2, 3, 5-7: "I would also like to see a bit more detail about what the lake and vegetation records from the Sahara suggest for the actual time period covered by the speleothems." Response: Although beginning to be recognised as a humid period elsewhere in the Mediterranean basin (Langgut et al., 2018), MIS3 is not well expressed in the Sahara region. Consequently, there is limited pollen or lake constraints to develop our understanding from. Generally, the Libyan interior is considered arid or hyperarid throughout the last glaciation (Cancellieri E. et al., 2016). Recent re-evaluation of lake levels in southwest Egypt indicates a groundwater fed system was active around 41

ka (Nicoll, 2018), which is similar to dates for springline tufa systems at Kharga Oasis (Smith et al., 2007). We are not aware of continental MIS3 pollen records from the region, but marine pollen from Tunisia indicates more arid conditions through the last glacial than during the Holocene (Brun, 1991). There is a triple peak in runoff from the Nile recorded in the marine sediment record, with maxima at 60, 55 and 35ka, indicating higher rainfall within the upper Nile catchment (Revel et al., 2010). We will include a summary of this evidence in the introduction, to provide better context for our new data.

C2, 3, 10-12: "A short discussion of the effects of different boundary conditions and how it could affect northern African moisture should be included." Response: We agree this would be useful, although keeping it short is a challenge for this complex system! Very briefly, the boundary conditions on northern African atmospheric moisture supply are 1) the sea surface temperature of the Atlantic and Mediterranean, 2) the surface water ïĄđ'18O of the same ocean regions, 3) land surface temperature of Africa and to a lesser extent southern Europe, 4) insolation (especially with respect to ITCZ position) and 5) the zonal pressure gradient across northern Africa.

C3, 1, 12-25: "I also think there should be a section in the introduction about the present day rainfall systems in the region. . . . . . . . . . . [error in citation of Celle-Jeanton et al 2001]. . . . . . . . . . . ..How were they [the rainfall end members] defined?........How were the averages in Figure 9 calculated?" Response: First – apologies for the error in the citation – the reference given in the References section is correct. We agree that including more detail about the modern rainfall system in the Introduction would be helpful. We also agree that we can improve description and definition of the end members. The Bet Dagan and Tunisian datasets we use are shown in Figure 5a, and do indeed have different meteoric water lines and D-excess characteristics. The sub-categories of rainfall within the Sfax dataset occupy different positions on the same meteoric water line (Celle-Jeanton et al., 2001). We have checked, and confirm that the same is true for the Bet Dagan data. The moisture sources used for the Bet Dagan site

is taken from the regional meteorology (Black et al., 2010;Gat et al., 2003). Moisture sources for Tunisia are as defined by Celle-Jeanton et al (2001), and we refer readers to this primary reference. Averages are arithmetic means, as this better reflects behaviour over time within systems where water throughput in and out of the karst system is likely very rapid compared to speleothem growth – we do not expect to be able to resolve synoptic events in this record.

C3, 1, 25-27: "What synoptic processes were involved in the formation of rain clouds – convection, advection? What circumstances lead to convective rainfall in the region?" Response: Convective systems, cyclones, upper-level troughs and static instabilities can all drive rainfall patterns in the Mediterranean basin and these modes are reviewed in (Dayan et al., 2015). Convection essentially reflects the relatively high SST of the Mediterrnaean during the winter, but rising air masses generally also need significant advection of moisture to drive significant rainfall. Upper level troughs reflect large-scale circulation (e.g. Red Sea Trough) or reflect lee effects downstream of mountains in the western Mediterranean, and promote rainfall in their regions of formation. The dominant cyclogenic centre is in the Gulf of Genoa, and secondary centres are placed in south Italy, Crete and Cyprus. Cyclonic systems can also penetrate from the Atlantic, where the high SST of the winter Mediterranean tends to sustain and amplify them, in close analogy to convection forcing. The key static instability is the penetration of the tropical air mass into the subtropical Mediterranean, forming a 'Saharan Cloud Band' at middle and upper atmospheric levels. These originate from within the ITCZ. As Libya is very sparsely instrumented, there is no literature we can find to specifically identify the synoptic processes involved in cloud formation precisely over our site. However, the Levant region is very well instrumented. Here, most rainfall falls under winter, low pressure conditions, and is convective (Peleg and Morin, 2012). The responsible low pressure systems can relate to transient, shallow lows over northern Israel, or less frequently more long-lasting Cyprus Lows or Red Sea Trough systems (Peleg and Morin, 2012).

C3, 1, 27-29 continuing to C4, 1, 1-2: "Further adding to the point above, it seems that the discussion of factors influencing stable isotopic reconstruction of rainfall is focussed on only the effects of rainfall amounts, temperature and moisture source. These are important factors, but I think one important factor is missing and that is the effect of cloud formation processes on the stable isotopic composition of rainfall". And C4, 1, 9-11: "The cloud formation processes of these winter storms and convection will affect the isotopic composition of the resulting rainfall......." Response: This is indeed an aspect we neglect, and deserves some further discussion. The high level of agreement between the absolute values of the fluid inclusion data and modern precipitation isotope data make it likely that similar condensation process are responsible for the MIS3 rainfall as are responsible today, making source effects the first order control on composition. Moreover, the modern precipitation and meteoric water lines derived from them already encompass the range of different condensation styles found in the modern Mediterranean. There are undoubtedly considerable further advances to be made from northern African speleothem fluid inclusion research, and we expect these nuances to be delineated by these future studies.

C4, 2, 3-7: "I think the arguments made for the mixing of the different end members would be much more clear if the discussion of the stable isotopic composition and the d-excess would be combined including clear definitions of the values for the three depositional phases of the speleothem and the modern rainfall end members". Response: If the Reviewer feels this will make our work more accessible, we will be very happy to follow their guidance.

C4, 3, 1-5 continuing to C5, 1, 1-7: "There seems to be a discrepancy between some of the statements in the manuscript with regards to the Atlantic rainfall source........... I think this needs to be clarified......... First it is stated that increased convection during phases with a low precession parameter must be related to a northward shift of the ITCZ, then the convection is attributed to enhance internal convection." Response: We do not see the core discrepancy that troubles the Reviewer on this point. We

can provide little positive evidence for Atlantic-sourced water in our record, and it is likely that other sites on the Mediterranean margin may show the same. Equally, it is undeniably true that to alter the Mediterranean freshwater budget the water must be external – and both the winter westerlies and the monsoon source much of that water from the Atlantic. We conclude that the key water responsible for Mediterranean freshening is likely to be arriving as runoff, not direct rainfall (which is not controversial – (Grant et al., 2016)). Primarily, we are aiming to warn colleagues working in the Mediterranean terrestrial sphere that their evidence of wet / dry changes may not relate directly to fresh / salty conditions in the Mediterranean Sea. Hopefully, this will become clearer with the restructuring this Reviewer recommends. The suggestion that recycled water from the Sahara could be important is interesting. Water re-evaporated from rainfall in northern Africa should be isotopically light, reflecting this relatively depleted source. We do not see a population of depleted fluid inclusions that sit outside of the modern rainfall system that would suggest there is a substantial contribution from such a source. C5, 4, 1-3: "The last paragraph of the section starting in line 123 is not really about 'the central North African speleothem record'. . . . . . and should maybe be in its own chapter". Response: agreed.

C5, 6, 1-3 (repeated in C5, 9, 1-2): "The speleothems carbonate stable isotopic composition were published, so the sentence. . . . . ... can be removed" Response: Agreed.

C5, 7,1-3: "I think this section should include a clear definition of the three depositional phases, giving the range of fluid inclusion stable isotopes and d-excess. This would make the comparison much easier." Response: Agreed.

C5, 11, 1-2 continued to C6, 1, 1-8: "Technical corrections". Response: Agreed – these changes should be made.

C6 continued into C7: "Figures" (presentational and formatting considerations for figures1, 4, 5, 6, 7 and 8). Response: Agreed – these changes should be made.

References

Black, E., Brayshaw, D. J., and Rambeau, C. M. C.: Past, present and future precipitation in the Middle East: Insights from models and observations, Philosophical Transactions of the Royal Society A: Mathematical, Physical and Engineering Sciences, 368, 5173-5184, 10.1098/rsta.2010.0199, 2010.

Brun, A.: Reflections on the pluvial and arid periods of the Upper Pleistocene and of the Holocene in Tunisia, Palaeoecology of Africa and the surrounding islands. Vol. 22. Proc. symposium on African palynology, Rabat, 1989, 157-170, 1991.

Cancellieri E., Cremaschi M., Zerboni A., and S., d. L.: Climate, Environment, and Population Dynamics in Pleistocene Sahara, in: Africa from MIS 6-2. Vertebrate Paleobiology and Paleoanthropology. , edited by: Jones S., and B., S., Springer, Dordrecht, 2016.

Celle-Jeanton, H., Zouari, K., Travi, Y., and Daoud, A.: Caractérisation isotopique des pluies en Tunisie. Essai de typologie dans la région de Sfax, Sciences de la Terre et des planètes, 333, 625-631, 2001. Dayan, U., Nissen, K., and Ulbrich, U.: Review Article: Atmospheric conditions inducing extreme precipitation over the eastern and western Mediterranean, Nat. Hazards Earth Syst. Sci., 15, 2525-2544, 10.5194/nhess-15-2525-2015, 2015.

Gat, J. R., Klein, B., Kushnir, Y., Roether, W., Wernli, H., Yam, R., and Shemesh, A.: Isotope composition of air moisture over the Mediterranean Sea: An index of the air-sea interaction pattern, Tellus, Series B: Chemical and Physical Meteorology, 55, 953-965, 10.1034/j.1600-0889.2003.00081.x, 2003.

Grant, K. M., Grimm, R., Mikolajewicz, U., Marino, G., Ziegler, M., and Rohling, E. J.: The timing of Mediterranean sapropel deposition relative to insolation, sea-level and African monsoon changes, Quaternary Science Reviews, 140, 125-141, http://dx.doi.org/10.1016/j.quascirev.2016.03.026, 2016.

Langgut, D., Almogi-Labin, A., Bar-Matthews, M., Pickarski, N., and Weinstein-Evron,

M.: Evidence for a humid interval at âĽij56–44 ka in the Levant and its potential link to modern humans dispersal out of Africa, Journal of Human Evolution, 124, 75-90, 10.1016/j.jhevol.2018.08.002, 2018.

Nicoll, K.: A revised chronology for Pleistocene paleolakes and Middle Stone Age – Middle Paleolithic cultural activity at Bîr Tirfawi – Bîr Sahara in the Egyptian Sahara, Quaternary International, 463, 18-28, https://doi.org/10.1016/j.quaint.2016.08.037, 2018.

Peleg, N., and Morin, E.: Convective rain cells: Radar-derived spatiotemporal characteristics and synoptic patterns over the eastern Mediterranean, J. Geophys. Res. D Atmos., 117, 10.1029/2011JD017353, 2012.

Revel, M., Ducassou, E., Grousset, F. E., Bernasconi, S. M., Migeon, S., Revillon, S., Mascle, J., Murat, A., Zaragosi, S., and Bosch, D.: 100,000 Years of African monsoon variability recorded in sediments of the Nile margin, Quaternary Science Reviews, 29, 1342-1362, 10.1016/j.quascirev.2010.02.006, 2010.

Rohling, E. J.: Environmental control on Mediterranean salinity and delta O-18, Paleoceanography, 14, 706-715, 1999.

Sharp, Z.: Principles of stable isotope geochemistry, 2017.

Smith, J. R., Hawkins, A. L., Asmerom, Y., Polyak, V., and Giegengack, R.: New age constraints on the Middle Stone Age occupations of Kharga Oasis, Western Desert, Egypt, Journal of Human Evolution, 52, 690-701, 2007.

Toucanne, S., Angue Minto'o, C. M., Fontanier, C., Bassetti, M.-A., Jorry, S. J., and Jouet, G.: Tracking rainfall in the northern Mediterranean borderlands during sapropel deposition, Quaternary Science Reviews, 129, 178-195, http://dx.doi.org/10.1016/j.quascirev.2015.10.016, 2015.

---

## Author Comment (AC2) · 29 Jan 2019

We thank the two anonymous Reviewers of our draft manuscript for their detailed and constructive reviews, and are extremely pleased that they find our work both interesting and worthy of publication. We fully concur that the interpretation of the data we present is complicated by structure of the dataset, and we are happy both reviewers agree with us that the data itself is so unique as to make a pressing case for publication and, that our analysis of it is fair, balanced and reasonable.

Below, we respond to the comments in order. The location code given for each comment and response represents the page the comment occurs in, followed by paragraph and lines (C, , -). Anonymous Reviewer 2

[Figure]

C2, 2, 6-10: "What I would like to see in the ms is an assessment of the extent to which the fluid inclusion isotope data are in isotopic equilibrium with the calcite. . . . . . . . . . . . . . . While isotope equilibrium is not a given for many speleothems, at least some consistency is to be expected between d18Occ and d18Ofi." Response: We concur with the reviewer's sentiment that difficulties in the correlation of the time-series are "uncomfortable", and this is why we approach the data by analysing large groups of datapoints rather than using a time-series approach. We are extremely pleased that this reviewer also feels that collection of the fluid inclusion dataset is "technically sound" (C1, 2, 7), and that the general accuracy of the dataset is supported by our quantitative analysis of it compared to modern rainfall isotopes (C2, 1, 1-2). It can only be concluded that these data are a good representation of this isotopic system, even though they look unusual.

We suggest that the apparently poor correlation of time-series likely arises from aliasing of a complicated signal in the fluid inclusions, and emphasise that it is unlikely the data presented are sufficiently resolved to demonstrate the two datasets actually have different structure. The correlation of the datasets is therefore ambiguous, rather than disproven. Sadly, the at least order-of-magnitude increase in the size of the fluid inclusion dataset needed to resolve this point is not realistic: indeed, this Reviewer notes that this dataset is already "comparatively large" (C1, 2, 1). The most appropriate way forward in this situation is to minimise the interpretation of the temporal structure of the fluid inclusion data we present, and this is what we have done. We are pleased that despite their discomfort, this Reviewer supports publication of this "interesting study" (C3, 3, 13-17).

A conventional equilibrium test is difficult to perform for this dataset, as each measurement comprises a mixture of inclusions with different compositions from each layer, and therefore an unknown position on the mixing line between these end members. Should we compute mean values (arithmetic or volume weighted); or extrapolate end members, and test for equilibrium of both? All these judgements require assumptions

we are not in a position to make. Consequently, we are only able to test for equilibrium in the subset of samples where the end members are sufficiently close together for the analysis to be fully "duplicable" (the subset shown in Figure 6). Modern mean winter temperature in Dernah (the nearest city to Susah Cave) is 11.9oC, with maximum 17.7oC and minimum 7.1oC. These fluid inclusions are therefore certainly at least close to isotopic equilibrium with the carbonate hosting them.

Table R1 (see attached)

C2, 2, 17-19: "It would perhaps be useful if the authors discuss that a bit more in the context of the interpretation of their record". Response: Agreed.

C3, 1, 5-10: "Bringing a third water source in, as is suggested in the ms, cannot really be supported by the data from my perspective. . . . . . . . . . One could perhaps argue that slight isotope changes within each of these moisture sources can cause similar isotope patterns?" Response:, The Reviewer agrees with our analysis that the data does show a mixing pattern of western and eastern Mediterranean sources (C3, 1, 4-5). So, we assume the 'third source' mentioned above is therefore the Atlantic external water we argue for, and find in relatively small amounts. We happily agree this is the most speculative part of our analysis. However, we also note that finding no Atlantic moisture at all in this dataset is a rather more startling interpretation than our suggestion that we find only a little. Atlantic-sourced moisture contributes to rainfall in central northern Africa today, and this mode of rainfall has previously been argued to be greater in past humid phases (Toucanne et al., 2015). We therefore find this point of speculation actually rather conservative in its nature.

C4, 2, 1-2: "I'd like to know where your duplicable samples from Fig. 6 are located in the stalagmite (stratigraphically). All in one period, or distributed all over?" Response: See Table 1. All three Growth Phases are represented by at least one fully duplicated sample.

C4, 2, 2-4: "Do you have a better correlation with the d18O values of the carbonate

when you consider the duplicable dataset only?" Response: Beyond the differences between the three phases (see next response), it is difficult to judge whether there is true correlation between the reduced fluid inclusion dataset and the calcite isotope dataset, because the former is rather small. To make interpretations based on such a "correlation" would seem to us rather speculative. We are safer limiting the discussion of the time series.

C4, 2, 4-5: "Further, I'd like to see if, based on the duplicable set only, one can still observe clear differences between the three wet intervals." Response: The sample from Phase II is more depleted both in ïĄď18Ofi and ïĄď18Occ than any of the samples from Phases I and III, which show similar compositions. Our interpretation that the water driving this middle growth phase is different to the other two is therefore supported by this additional analysis.

C4 paragraphs 3, 4 and 5: "Figures" Response: See response to Reviewer 1.

C4, 6, 1-2: Error in text. Response: Correction will be made.

C4, 7, 1-5: "It would be interesting to know. . . . . . . . . . . could you have any sea spray effect?" Response: Given that we find no clear signal (or indeed, no variance beyond measurement uncertainty) in the Sr isotope record, these points cannot alter the interpretation and we are consequently unclear about their relevance (?).

C4, 8, 1-2 continued to C5, 1, 1-5: "Towards the end of the discussion, d13CC plays an important role. These data, however, are not shown. . . . . . . . .Shouldn't d18Ofi do the same as d2Hfi if your claim is correct?" Response: ïĄď13Ccc is actually shown, within Figure 8b. We do make greater use of ïĄď2Hfi towards the end of the discussion, because we are attempting to use the fluid inclusion data to better understand the carbonate isotope datasets, and hydrogen cannot be measured in carbonate and therefore provides valuable independent evidence of changes. As the fluid inclusion isotopes do correlate, it is indeed true that the ïĄď18Ofi shows a similar pattern.

C5, 2, 1: "Your statement in line 392 to 394 is not clear to me. Response: We will clarify that statement.

References

Sharp, Z.: Principles of stable isotope geochemistry, 2017.

Toucanne, S., Angue Minto'o, C. M., Fontanier, C., Bassetti, M.-A., Jorry, S. J., and Jouet, G.: Tracking rainfall in the northern Mediterranean borderlands during sapropel deposition, Quaternary Science Reviews, 129, 178-195, http://dx.doi.org/10.1016/j.quascirev.2015.10.016, 2015.

—————————————————

Table R1

| Duplicated Sample | Mean $\delta^{18}O_{fi}$ | Mean $\delta^2H_{fi}$ | Distance From Base (mm) | Age | $\delta^{18}O_{cc}$ | Temperature (°C)[1] | Growth Phase |
|---|---|---|---|---|---|---|---|
| 9 | -4.35 | -21.51 | 15-20 | 66394-66388 | -3.78 | 13.3 | I |
| 11 | -5.69 | -19.95 | 56-61 | 66493-64393 | -3.96 | 8.7 | I |
| 18 | -5.92 | -21.75 | 109-118 | 63487-63302 | -3.98 | 7.9 | I |
| 23 | -5.45 | -25.62 | 260-267 | 59106-58490 | -4.51 | 11.8 | I |
| 25 | -5.38 | -25.54 | 395-400 | 51687-51648 | -5.21 | 15.0 | II |
| 33 | -4.70 | -26.72 | 556-561 | 37260-37221 | -4.37 | 14.3 | III |
| 29 | -5.08 | -22.72 | 597-602 | 36957-36921 | -4.6 | 13.7 | III |
| 35 | -5.03 | -21.37 | 652-657 | 36581-36284 | -3.82 | 10.8 | III |
| 7 | -4.96 | -23.74 | 767-772 | 35688-35647 | -4.32 | 13.1 | III |

[1] Calculated using the equation presented by (Sharp, 2017).

**Fig. 1.** Table R1

---

## Referee Report (RR1)

This manuscript presents a new data set of fluid inclusion stable isotopes in a speleothems from northeastern Libya. The speleothem grew intermittently from late MIS 4 through late MIS 3 with three major deposition phases (Phases I-III) which are reflecting with humid conditions in the now arid region. While Phases I and III are associated with high northern hemisphere insolation caused by high precession, Phase II humidity is probably caused by obliquity cycles. The stable isotopic compositions of the fluid inclusions and the d-excess parameter related to them suggest a change in the moisture sources associated with the different orbital forcing with Phases I and III having mixed moisture sources in the eastern and western Mediterranean, whereas Phase II mainly received western Mediterranean moisture.

I think that the paper addresses a relevant scientific topic within the scope of CP and it presents important new data for the northern African region with substantial conclusions. I think the scientific methods are valid and sufficiently described. The results do support the conclusions and the authors give proper credit to previous work. I think the title and Abstract are clear and concise and the overall presentation is well-structured. Language is fluent and precise. I don't think that major clarifications or restructuring of the manuscript will be needed. The number and quality of the references is appropriate and so is the Supplementary Material.

Furthermore the authors responded appropriately to previous comments on this manuscript and I only have few technical corrections which will be listed below.

Line 80: "…this primarily affected the northern Mediterranean margin only…" – I think either "primarily" or "only" should be cut here.

Line 213: "Fluid inclusions for Phases I and II…." Should be Phases I and III.

Line 324: "…and why during some periods in Susah Cave show strong correlation…", delete "during".

Line 364-366: "At Sfax today, this influence causes a prominent bimodal behavior…, which eliminates a simple and quantitative rainfall amount control on precipitation, which can be observed at Tunis." Two relative clauses with which in one sentence, could probably be rephrased.

Line 373: :…, but this is too complicated by independent changes increased …" delete "changes".

Line 413-414: "…,but it not consistent with …" should be "but is not consistent with"?

Comment on the Supplementary Material:

The word diagenetic occurs several times in the Supplementary text in quotation marks. If the authors do not believe that processes affecting the fluid inclusions could be diagenetic, then I think the word should be removed/replaced (e.g. with alteration). If the processes could be diagenetic, then the quotation marks are not needed.

---

## Referee Report (RR2)

Review of the 1st revision of "Enhanced Mediterranean water cycle explains increased humidity during MIS 3 in North Africa"

Based on the revised ms, and the answers by Rogerson et al to my comments and questions on the initial submission, I conclude that I can now support final publication of this ms in COTP.

Please check lines 155, 324-325, 373-374, 413-414, where I believe I spotted some typos/grammar inconsistencies.

---

## Author Response (AR2)

**Enhanced Mediterranean water cycle explains increased humidity during MIS 3 in North Africa - Second Response to reviewers**

Many thanks again to the two anonymous Reviewers, the Editor and Journal Staff for their help with this manuscript. We have made very change recommended by the Reviewers in the new version of the manuscript.

Best wishes,

Mike